# Evaluation of the 8th Edition AJCC Staging System for the Clinical Staging of Pancreatic Cancer

**DOI:** 10.3390/cancers14194672

**Published:** 2022-09-26

**Authors:** Huapyong Kang, Seung-seob Kim, Min Je Sung, Jung Hyun Jo, Hee Seung Lee, Moon Jae Chung, Jeong Youp Park, Seung Woo Park, Si Young Song, Mi-Suk Park, Seungmin Bang

**Affiliations:** 1Department of Internal Medicine, Gil Medical Center, Gachon University College of Medicine, Incheon 21565, Korea; 2Department of Medicine, Yonsei University Graduate School, Seoul 03722, Korea; 3Department of Radiology, Yonsei University College of Medicine, Seoul 03722, Korea; 4Digestive Disease Center, CHA Bundang Medical Center, CHA University School of Medicine, Seongnam 13496, Korea; 5Division of Gastroenterology, Department of Internal Medicine, Yonsei University College of Medicine, Seoul 03722, Korea

**Keywords:** pancreatic neoplasms, adenocarcinoma, neoplasm staging, prognosis, lymph node

## Abstract

**Simple Summary:**

The 8th edition of the AJCC staging system for pancreatic cancer has been validated for pathological staging; however, its suitability for clinical staging is still uncertain. Clinical staging is important for pancreatic cancer because surgical resection is not suitable for most patients. We validated the prognostic performance and suitability of the current staging system for clinical staging. In our analysis using a prospectively collected pancreatic cancer cohort database, survival difference was not shown between stage IB and IIA, which were divided by the new size-based T3 criterion. Among the patients who received surgery, the pathological stage was more advanced than the clinical stage in 57.3%, mostly due to a false-negative lymph node in clinical staging. These findings suggest that the 8th edition AJCC staging system is not validated for clinical staging and separate criteria more suitable for clinical staging should be established.

**Abstract:**

The 8th edition of the American Joint Committee on Cancer (AJCC) staging system for pancreatic cancer (PC) has been validated for pathological staging; however, its significance for clinical staging remains uncertain. We validated the prognostic performance and suitability of the current staging system for the clinical staging of PC. We identified 1043 patients from our PC registry who were staged by imaging according to the 8th edition staging system and conducted analysis, including overall survival (OS) comparison. Gradual prognostic stratification according to stage hierarchy yielded significant OS differences between stage groups, except between stage I and II (*p* = 0.193). A substage comparison revealed no survival differences between IB (T2N0) and IIA (T3N0), which were divided by the T3 criterion only (*p* = 0.278). A higher N stage had significantly shorter OS than a lower N stage (all pairwise *p* < 0.05). However, among the 150 patients who received upfront surgery, the pathological stage was more advanced than the clinical stage in 86 (57.3%), mostly due to a false-negative cN0 (70.9%). Our results suggest that the new definition of T3 and the number-based N criteria in the 8th edition AJCC staging system may be not adequate for clinical staging. Establishing separate criteria more suitable for clinical staging should be considered.

## 1. Introduction

The American Joint Committee on Cancer (AJCC) staging system, based on the primary tumor, regional lymph nodes (LNs), and metastasis (TNM), is a universally accepted tool for cancer staging. From initial diagnosis, it allows for disease-extent determination, appropriate treatment-approach selection, and prognosis prediction [1]. In 2016, the AJCC released its 8th edition staging system with several updates in TNM staging [2]. The staging system for pancreatic cancer (PC), one of the most lethal malignant diseases globally [3,4,5], was also changed based on the evidence that tumor size and the number of positive LNs (PLNs) were significantly related to prognosis [6,7,8,9]. Therefore, the T3 criterion was revised as a tumor size > 4 cm, and the PLN number-based N criteria, including N1 (1–3 PLNs) and N2 (≥4 PLNs), were introduced. After criteria revision, this staging system was validated and compared with the former staging system in several studies with data from patients with surgically resected PC [10,11,12,13]. Overall, these studies concluded that pathological staging with the 8th edition AJCC staging system was valid and had comparable or better discriminating power than the previous edition.

However, the current 8th edition staging system for PC was established and revised based on pathological data from patients with surgically resected PC exclusively, and its applicability for clinical staging remains unclear [14]. Notably, more than 80% of patients diagnosed with PC are not suitable for curative resection [15], and clinical staging is the only method of post-diagnostic implementable classification in most patients with PC. Therefore, it is important to evaluate whether the 8th edition AJCC staging system is appropriate for clinical staging in anticipation of an upcoming revision of the staging system; however, data are still lacking.

Herein, we validated the prognostic performance and suitability of the 8th edition AJCC staging system for the clinical staging of PC.

## 2. Materials and Methods

### 2.1. Patient Selection

We analyzed data from the Severance Hospital Pancreatic Cancer Cohort Registry. This cohort registry is a prospectively collected database of patients with PC treated at Severance Hospital since 2015. Patient information regarding demographic, clinical, radiographic, pathologic, and treatment characteristics were collected, as well as follow-up and survival details. Data from patients who were diagnosed with PC from 2012 to 2014 were retrospectively collected after obtaining study approval. We included patients diagnosed with PC from January 2012 to April 2017 aged > 19 years with histologically or cytologically proven pancreatic malignancy. We excluded patients with confirmed extrapancreatic malignancy and a confirmed pancreatic neuroendocrine tumor or carcinoma because different staging systems are used for these patients. We also excluded patients with inadequate imaging data with which to conduct clinical staging. Basic demographics, clinical information at initial diagnosis, pathological information after resection, and the date of the last follow-up or death were extracted for analysis. This cohort study was approved by the Institutional Review Board of Yonsei University Health System (Approval number: 4-2015-1058) and conducted in accordance with the Declaration of Helsinki.

### 2.2. Imaging Studies for Clinical Staging

All patients underwent multidetector computed tomography (MDCT) and magnetic resonance imaging (MRI) at initial diagnosis. Five board-certified abdominal radiologists with 5–20 years of experience analyzed the imaging findings and entered them into the cohort database. They recorded the tumor size and location and evaluated the major vessels (celiac artery, superior mesenteric artery, common hepatic artery, portal vein [PV], and superior mesenteric vein [SMV]). When >180° of a vessel’s circumference was in contact with the cancer, or a vessel contour showed deformity, the concerned vessel was regarded as invaded [16]. When the outer margin of the pancreas was bulging with a loss of its lobulated contour, it was regarded as an extrapancreatic extension of the tumor. LNs were also analyzed, and the number of radiographic PLNs was counted. A radiographic PLN was defined as an LN with a short axis longer than 1 cm or showing more than two features of the following: round rather than ovoid contour, heterogeneity, and central necrosis [16]. Conglomerated, but distinct, LNs were counted separately. Finally, the presence of distant metastasis was recorded. Moreover, we used available information from endoscopic ultrasonography or positron emission tomography (PET) for clinical staging. All imaging study data collected in the cohort database were used for clinical staging according to the 7th and 8th editions of the AJCC staging system (Table 1). All patients were initially staged according to the 7th edition AJCC staging system at the time of their registration to the cohort. When the 8th edition staging system was introduced in early 2017, all previously registered patients were restaged according to the 8th edition staging system with newly added information on the radiographic PLN number.

### 2.3. Statistical Analysis

For the analysis of survival data, the Kaplan–Meier method was used to estimate median survival with 95% confidence intervals (CI), and pairwise comparisons between each curve were performed using the log-rank test. Overall survival (OS) was defined as the period spanning the date of the reference imaging study for clinical staging to the date of death or the last follow-up. The cut-off date of follow-up for this study was 28 July 2020. Patients alive at the end of follow-up were censored from survival analysis. The performances of both the 8th and 7th editions of the AJCC staging systems were compared in three aspects: discrimination ability, homogeneity, and monotonicity of the gradient. For the evaluation of discrimination ability, Harrell’s concordance index (C-index), Heagerty’s integrated area under the curve (iAUC), and the Akaike information criterion (AIC) were calculated. The C-index and iAUC have values ranging between 0 and 1.0, with values closer to 1.0 indicating more significant discriminating abilities [17,18]. A smaller AIC value indicates a better model for prognostic stratification [19]. Homogeneity was measured using the likelihood ratio chi-square test (LR). Monotonicity of the gradient was analyzed by applying the linear trend chi-square test (LT). A Cox proportional hazards model was used to compute the C-index, iAUC, AIC, LR, and LT values. A *p* value < 0.05 was considered statistically significant. Descriptive statistics and survival analyses were performed with IBM SPSS (version 23.0, IBM Corp., Armonk, NY, USA), whereas performance parameters were computed using R package, version 3.4.3 (http://www.R-project.org accessed on 15 May 2018), and SAS (version 9.4, SAS Inc., Cary, NC, USA).

## 3. Results

### 3.1. Patient Characteristics

Of the 1162 patients registered in our cohort registry from January 2012 to April 2017, 1059 patients met the inclusion criteria. After excluding 16 patients (10 with confirmed neuroendocrine carcinoma, 4 with confirmed extrapancreatic malignancy, and 2 with inadequate imaging studies), 1043 patients were finally analyzed. The baseline characteristics of all selected patients are summarized in Table 2. Regarding treatment after diagnosis, 150 (14.4%) patients received upfront surgery, and 31 (3.0%) underwent resection after neoadjuvant therapy. Sixty-nine (6.6%) patients did not receive surgery after neoadjuvant therapy. Adjuvant therapy was given to 129 (12.4) patients, 719 (68.9%) received palliative treatment for initially unresectable or recurrent PC, and 188 (18.0%) patients received only best supportive care. Detailed information on neoadjuvant and adjuvant therapy is presented in Appendix A. Patient distribution according to the 8th edition AJCC staging system is summarized in Table 3.

### 3.2. Survival Analysis between Stage Groups under the 8th Edition Staging System for PC

The median duration of follow-up was 10.0 months (range, 0.3–87.4). Figure 1A shows the Kaplan–Meier plots of median OS and the result of pairwise comparison among the four stage groups according to the 8th edition staging system. We noted a gradual prognostic stratification, along with stage hierarchy, with significant OS differences between all stage groups, except between stage I and II (I 19.6 vs. II 14.7 months; *p* = 0.193). To analyze the statistical insignificance of survival differences between stage I and II, we compared patient OS for four substages of stages I and II (Figure 1B). In this analysis, the OS of stage IB and IIA patients was not different (IB 16.8 vs. IIA 19.2 months; *p* = 0.278). This result indicated that the OS of T2 did not differ from that of T3, which is newly defined as a tumor size > 4 cm in the 8th edition staging system, because stages IB and IIA are only defined by T stages. Statistical differences in OS were also not shown between stage IA/IIA and IB/IIB patients; nonetheless, there were strong trends toward longer patient OS in earlier stages (IA 35.8 vs. IIA 19.2 months, *p* = 0.094; IB 16.8 vs. IIB 10.1 months, *p* = 0.058). The same analysis performed in 942 patients with confirmed adenocarcinoma shows consistent results with those from all 1043 patients (Appendix A).

### 3.3. Subgroup Analysis in Patients with Non-Metastatic PC: Validation of Newly Defined N Stages

Survival analysis according to the N stage was performed to evaluate the validity of the new size-based N criteria of the 8th edition staging system in patients with non-metastatic PC (n = 504). In this analysis, only 13 patients (2.6%) with non-metastatic PC were classified as clinical N2, and 62 patients (12.3%) were classified as clinical N1. Figure 2 shows the Kaplan–Meier plots wherein higher N stages had shorter median OS than lower N stages, and these differences were confirmed in a pairwise comparison. The same analysis performed in 456 patients with confirmed adenocarcinoma shows consistent results with those presented above (Appendix A).

### 3.4. Subgroup Analysis of Patients Who Underwent Upfront Surgery: Clinical Staging Accuracy

In our study cohort, 150 patients underwent upfront surgery without neoadjuvant therapy. Table 4 shows the distribution of preoperative clinical stages and postoperative pathological stages. With reference to the pathological stage, the accuracy of clinical staging was 36% (54 of 150). Compared to their clinical stage, 86 of the 150 patients (57.3%) had a more advanced pathological stage, mostly due to a radiographic false negative diagnosis of PLN (61 of 86, 70.9%). The overall sensitivity, specificity, positive predictive, and negative predictive values of radiographically detected PLNs were 13.51%, 98.68%, 90.91%, and 53.96%, respectively.

### 3.5. Comparison of Prognostic Performance between the 8th and 7th Editions

Table 5 shows the comparison of prognostic performance between the current and previous editions of the AJCC staging system. The C-index and iAUC values were closer to 1.0 with the 8th edition (8th vs. 7th; 0.671 vs. 0.667; 0.654 vs. 0.650, respectively), with significant differences (0.004 [95% CI, 0.001–0008]; 0.004 [95% CI, 0.001–0.008], respectively), compared to the 7th edition. Furthermore, the AIC values were smaller (9878.689 vs. 9885.982) with the 8th edition. These results indicate that the 8th edition has more significant discriminating ability than the 7th edition staging system. The calculated LR was higher and the calculated LT was lower with the 8th edition than with the 7th edition (286.0384 vs. 278.7454; 195.7435 vs. 214.2208, respectively). These findings suggested that the 8th edition staging system was a better model than the 7th edition in terms of homogeneity, but not in terms of monotonicity of the gradient.

## 4. Discussion

The well-known drawback of the 7th edition staging system of PC was a shift in patient distribution to T3, which made T1 and T2 rare [13]. This can reduce the accuracy of prognostic stratification, particularly in earlier stages. However, this deviation was solved with the new 8th edition T-stage criteria, as demonstrated in previous studies conducted using pathological staging [10,12,13,20], and in the present study of the use of clinical staging as well. However, along with this change in the present study, the number of patients with stage I PC (T1 and T2) markedly increased, whereas the number of patients with stage II PC (T3 and N1) decreased to that of the smallest group (6.4%).

Survival analysis by the 8th edition in our study showed a gradual prognostic change along with the stage hierarchy, with a significant or strong trend toward having OS differences, except between stage IB (T2N0) and IIA (T3N0). This finding could put into question the validity of the revised T3 criterion in the 8th edition. While several studies evaluating patients with surgically resected PC had validated the 4 cm tumor size cut-off [6,10,12,13], a recent study by Li et al. with 1349 patients showed contradictory results. Similar to the results of our study, they showed no survival difference between pathologically staged IB and IIA according to the 8th edition [21]. For clinical staging, evidence is lacking on the applicability of the size-based T3 criterion so far. This is likely because the 4 cm tumor size cut-off was determined using data from resected PC cases only. In our study, over 95% of the patients with a tumor size > 4 cm had arterial invasion (T4) or distant metastasis (M1). Thus, unlike in pathological staging, stage IIA may merely be nominal in clinical staging, and this could imply a limited applicability of the sized-based T3 criterion in this setting. Our results showed several advantages for T classification of the 8th edition in terms of stage distribution and prognosis-discriminating ability over the 7th edition, similar to the results from a recent study on pathological staging [10]. At the same time, the necessity of a more suitable T classification for clinical staging is raised. It may be helpful to include factors constituting resectability criteria, given the prognostic impact of surgical resection. In this aspect, radiographic PV or SMV invasion, which can significantly affect the prognosis of patients with non-metastatic PC [22], may be worth considering as a factor in T classification.

The PLN number-based N staging of the 8th edition showed the ability to discriminate prognosis between N-stage groups in patients with non-metastatic PC in the present study. However, stage IIB was very rare, and only five patients were classified as stage III by N2 without T4. This clearly highlights the limitations of imaging modalities, wherein the accurate determination of radiographic PLN was very difficult. Several studies of pancreatobiliary cancer have shown unsatisfactory accuracy for MDCT, MRI, and PET in PLN detection. Overall reported sensitivity and negative predictive value ranges are 15–57% and 53–72%, respectively [23,24,25,26,27,28,29,30], indicating a high false negative rate. Similarly in our study, after upfront surgery in 150 patients, 57.3% had a more advanced pathological stage than that given at preoperative clinical staging, mostly due to a radiographic false negative PLN. Moreover, we demonstrated relatively low negative predictive values (53.96%) for imaging modalities in PLN detection. This result indicates the likelihood of missing many patients who should be classified as stage IIB or III. To increase the diagnostic accuracy of nodal staging in pancreatic cancer, several criteria have been evaluated, such as a combination of MDCT and serum CA 19-9 [26], lymph node size [31], and expanded morphological criteria [32]. To date, however, no specific criteria have shown satisfactory accuracy for nodal staging. While the number of PLN was suggested as a prognostic factor in patients with resected PC and the cut-off number of PLN for staging was validated in several previous studies [7,8,9,10,12,33,34,35], some studies have reported that applying the PLN number cut-off elicited no survival differences [13,20,36]. Considering the suboptimal accuracy of preoperative PLN detection, we suggest that N staging needs to be simplified as in the 7th edition, at least for clinical staging.

Despite some improvements in prognostic performance in the revised staging system for PC, it still cannot be used as a guide for clinical management, which is a major role of clinical staging [14]. For example, venous criteria are not included in the T classification, and therefore, patients with unresectable PC due to unreconstructible venous occlusion can even be classified as early stage PC according to the current staging system, thereby diminishing the prognostic significance. The separate use of clinical and pathological staging classifications as in current esophageal and gastric cancer staging systems will be an appropriate solution to this problem [37,38].

This study had several limitations. First, we analyzed cohort study data from a single institution. Second, post-diagnostic treatment and the clinical courses of each patient were heterogeneous. Therefore, our results do not reflect prognostic prediction after certain kinds of treatment. Third, we included patients with cancers other than adenocarcinoma, such as unspecified carcinoma and intraductal papillary mucinous carcinoma. Although such patients were few (9.7%), they could limit the pathology-specific interpretation of our result. Generally, with the evidence of highly suspected PC on imaging and serologic studies, patients with these histocytological results are diagnosed as having PC, and the treatment is the same as that of adenocarcinoma [39]. In addition, we excluded patients with neuroendocrine tumors whose staging system and treatment differed from those of patients with adenocarcinoma. Fourth, we did not compare the accuracy of clinical staging between specific imaging modality or criteria. Thus far, integrating information from all available imaging modalities is required to make a more accurate clinical staging than relying on a single modality. Fifth, we are unable to suggest better T and N classifications for clinical staging based on our data. The number of patients with resectable and borderline resectable PC might not be adequate to discriminate minor prognostic differences between stages. Further research for developing a more suitable classification for clinical staging is needed using larger multicenter or national cohorts. Despite these limitations, our study is meaningful in that we showed notable flaws in the current staging system for the clinical staging of PC. To the best of our knowledge, this is the first study with a detailed evaluation of the 8th edition AJCC staging system for PC in a pure clinical staging setting.

## 5. Conclusions

Our results suggest that the new definition of T3 in the 8th edition AJCC staging system may be not adequate for clinical staging. With the suboptimal accuracy of the current imaging system in PLN detection, the applicability of the new number-based N criteria remains controversial. Given that the clinical stage is the only assessable stage in most patients with PC, establishing more suitable criteria for clinical staging should be considered in upcoming revisions, separately from pathological staging criteria.

## Figures and Tables

**Figure 1 cancers-14-04672-f001:**
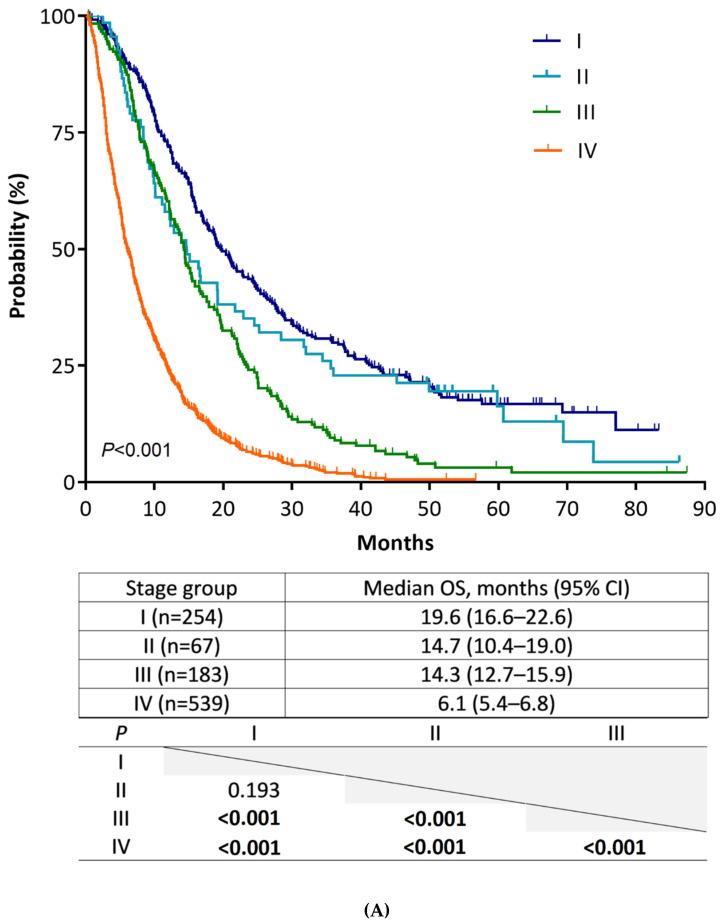
Kaplan–Meier estimates of overall survival of stage groups according to the AJCC staging system, 8th edition. (**A**) Comparison of stages (from I to IV). (**B**) Comparison of substages (from IA to IIB).

**Figure 2 cancers-14-04672-f002:**
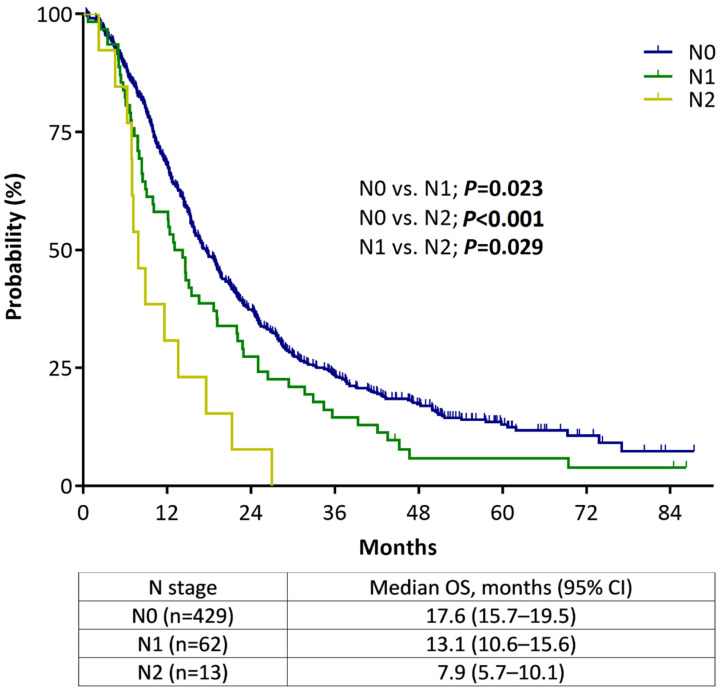
Overall survival of non-metastatic pancreatic cancer patients according to the AJCC staging system, 8th edition, stratified by N staging.

**Table 1 cancers-14-04672-t001:** American Joint Committee on Cancer 8th Edition Staging System for Pancreatic Cancer.

T	Primary Tumor	N	Number of Regional PLNs	Stage	T	N	M
T1	≤2 cm	N0	0	IA	1	0	0
T2	>2 cm, ≤4 cm	N1	1 to 3	IB	2	0	0
T3	>4 cm	N2	≥4	IIA	3	0	0
T4	CA, SMA, and/or CHA invasion			IIB	1–3	1	0
				III	4	Any	0
					Any	2	
				IV	Any	Any	1

Abbreviations: PLN—positive lymph node; CA—celiac axis; SMA—superior mesenteric artery; CHA—common hepatic artery.

**Table 2 cancers-14-04672-t002:** Patient Characteristics.

	n = 1043
Sex	
Female	466 (44.7)
Male	577 (55.3)
Age, years	
	65 (30–89)
BMI, kg/m^2^	
	22.5 (13.4–34.9)
ECOG-PS	
0	534/798 (66.9)
1	207/798 (26.0)
≥2	57/798 (7.1)
Level of CA 19-9	
Normal	239/1035 (23.1)
Elevated	796/1035 (76.9)
Tumor location in the pancreas	
Head	454 (43.5)
Body	217 (20.8)
Tail	192 (18.4)
Overlapped	180 (17.3)
Tumor size on image at diagnosis, cm	
	3.7 (0.2–15.8)
Resectability	
Resectable	205 (19.7)
Borderline resectable	112 (10.7)
Locally advanced	187 (17.9)
Metastatic	539 (51.7)
Pathologic diagnosis	
Adenocarcinoma	942 (90.3)
IPMN with invasive carcinoma	14 (1.3)
Unspecified carcinoma	87 (8.4)
Differentiation	
Well	94 (9.0)
Moderate	301 (28.9)
Poor	109 (10.5)
Not evaluable	539 (51.7)
Treatment	
Upfront surgery	150 (14.4)
Neoadjuvant therapy, resected	31 (3.0)
Neoadjuvant therapy, not resected	69 (6.6)
Adjuvant therapy	129 (12.4)
Palliative therapy (chemo or CCRT)	719 (68.9)
Only BSC	188 (18.0)

Abbreviations: BMI—body mass index; ECOG-PS—Eastern Cooperative Oncology Group performance status; CA—carbohydrate antigen; IPMN—intraductal papillary mucinous neoplasm; CCRT—concurrent chemoradiotherapy; BSC—best supportive care. Variables are presented as a median (range) or n (%).

**Table 3 cancers-14-04672-t003:** Patient Distributions of Clinical Stages.

**n = 1043**		**8th Edition**			
		**n**		**%**	
T Stage					
T1		81		7.8	
T2		364		34.9	
T3		184		17.6	
T4		414		39.7	
N stage					
N0		708		67.9	
N1		207		19.8	
N2		128		12.3	
Stage					
IA (T1N0M0)		70		6.7	
IB (T2N0M0)		184		17.6	
IIA (T3N0M0)		39		3.7	
IIB (T1–3N1M0)		28		2.7	
III (T4 or N2M0)		183		17.5	
IV (TxNxM1)		539		51.7	
**n (%)**		**8th edition**			
		**T1**	**T2**	**T3**	**T4**
All patients (n = 1043)	N0	76 (7.3)	285 (27.3)	101 (9.7)	246 (23.6)
	N1	4 (0.4)	58 (5.6)	50 (4.8)	95 (9.1)
	N2	1 (0.1)	21 (2.0)	33 (3.2)	73 (7.0)
Localized (M0) (n = 504)	N0	70 (13.9)	184 (36.5)	39 (7.7)	136 (27.0)
	N1	2 (0.4)	20 (4.0)	6 (1.2)	34 (6.7)
	N2	1 (0.2)	3 (0.6)	1 (0.2)	8 (1.6)

**Table 4 cancers-14-04672-t004:** Changes in Distribution of Patients before and after Upfront Surgery according to the AJCC 8th Edition (n = 150).

		Pathological Stage					
		IA	IB	IIA	IIB	III (All N2)	Total, n (%)
Clinical stage	IA	13	16	4	13	8	54 (36.0)
	IB	5	28	2	24	12	71 (47.3)
	IIA	0	0	7	3	1	11 (7.3)
	IIB	0	1	0	5	3	9 (6.0)
	III	0	0	0	4	1	5 (3.4)
Total, n (%)		18 (12.0)	45 (30.0)	13 (8.7)	49 (32.7)	25 (16.6)	150

**Table 5 cancers-14-04672-t005:** Comparison of Performance.

	C-Index (95% CI)	iAUC (95% CI)	AIC	Likelihood Ratio χ^2^	Linear Trend χ^2^
8th edition	0.671 (0.653–0.688)	0.654 (0.637–0.670)	9878.689	286.0384	195.7435
7th edition	0.667 (0.649–0.684)	0.650 (0.634–0.665)	9885.982	278.7454	214.2208
8th vs. 7th	0.004 (0.001–0.008)	0.004 (0.001–0.008)	-	-	-

Abbreviations: C-index, Harrell’s concordance index; CI—confidence interval; iAUC—Heagerty’s integrated area under the curve; AIC—Akaike information criterion.

## Data Availability

The data presented in this study are available in this article and Appendix A.

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
