# Peer review of "Evaluation of the 8th Edition AJCC Staging System for the Clinical Staging of Pancreatic Cancer"

_cancers, 2022, doi:10.3390/cancers14194672_

Round 1
Reviewer 1 Report
To my opinion the manuscript is well-written, the study is well-conducted, and the methodology is correct
I also agree with the conclusions that the authors present
The only weakness of the study might be the inclusion of locally advanced and metastatic patients in a study that examines the diagnostic and prognostic accuracy of TN stage. Practically, the population of interest is 205 pts with resectable and 112 pts with borderline resectable disease. Therefore, the size of the study population of interest might not be adequate to discriminate potential minor differences in prognosis between distinct stages.
Nevertheless, I agree that the current radiological methods are not adequate to provide a specific estimation of tumor clinical stage, especially N stage. In contrast, radiological T stage is more accurate, but I agree with the authors that the inclusion of the presence or absence of vein involvement might improve its prognostic accuracy.
A larger study might be able to answer the above questions
Author Response
All line and page numbers indicating changes based on revised manuscript with track changes
Reviewer #1.
To my opinion the manuscript is well-written, the study is well-conducted, and the methodology is correct
I also agree with the conclusions that the authors present
The only weakness of the study might be the inclusion of locally advanced and metastatic patients in a study that examines the diagnostic and prognostic accuracy of TN stage. Practically, the population of interest is 205 pts with resectable and 112 pts with borderline resectable disease. Therefore, the size of the study population of interest might not be adequate to discriminate potential minor differences in prognosis between distinct stages.
Nevertheless, I agree that the current radiological methods are not adequate to provide a specific estimation of tumor clinical stage, especially N stage. In contrast, radiological T stage is more accurate, but I agree with the authors that the inclusion of the presence or absence of vein involvement might improve its prognostic accuracy.
A larger study might be able to answer the above questions
Answer:
Thank you for taking your valuable time to review our manuscript.
We agree with your comment on the number of patients with resectable and borderline resectable pancreatic cancer (RPC and BRPC) in our study. Still, most of pancreatic cancer is diagnosed at advanced stage and localized early stages take minor proportion. Therefore, collecting clinical information of larger number of patients with RPC and BRPC than our study may require multicentric effort. We also agree with you that a larger multicenter study could validate the results from our study and suggest better clinical staging system. We have added this in the discussion section as a limitation of our study.
Revision:
Line 305 on Page 10
“The number of patients with resectable and borderline resectable PC might not be adequate to discriminate minor prognostic differences between stages.”
Reviewer 2 Report
This is a well written manuscript where the authors used their own data base to evaluate how changes to the most recent AJCC manual influenced outcomes compared to previous versions.
1. The findings suggest that clinical staging does not accurately predict pathologic staging and additional criteria are needed such as inclusion of ca 19-9. Can the authors comment on whether they saw differences between CT and MRI in their cohort as it relates to sensitivity/specificity of nodal involvement?
2. The authors comment that pathologic size was not an accurate predictor of OS. The suggest that adding information related to venous involvement may be more predictable. Is it possible to include this data from their database?
3. The manuscript would be stronger if they excluded other types of cancer such as IPMN. Although they claim the prevalence is small, they should do a sensitivity analysis to confirm that it minimally influences their outcomes.
4. You report that adjuvant therapy was given in 129 (12.4) patients. How is "adjuvant" therapy defined? Is it only after surgery or does it include those who received neoadjuvant therapy?
5. Can you include more information on what therapy patients received both in the neo-adjuvant and adjuvant settings?
Author Response
All line and page numbers indicating changes based on revised manuscript with track changes
Reviewer #2
This is a well written manuscript where the authors used their own data base to evaluate how changes to the most recent AJCC manual influenced outcomes compared to previous versions.
- The findings suggest that clinical staging does not accurately predict pathologic staging and additional criteria are needed such as inclusion of ca 19-9. Can the authors comment on whether they saw differences between CT and MRI in their cohort as it relates to sensitivity/specificity of nodal involvement?
Answer:
Thank you for taking your valuable time to review our manuscript.
In our pancreatic cancer cohort database, clinical staging was recorded based on integrated information from all performed imaging studies including CT, MRI, PET-CT and EUS. Therefore, it is not possible to compare diagnostic performance for nodal staging between CT and MRI using our database.
Previous studies evaluating diagnostic performance of detecting lymph node metastasis in patient with pancreatobiliary cancer showed 37.0-57.0% of sensitivity and 76.1-100% of specificity using MDCT, and 15.0-58.3% of sensitivity and 83.3-93.0% of specificity using MRI.1-4 Comparative studies did not show differences in nodal diagnostic performance between CT and MRI.1,5
Regarding these suboptimal performances for detecting positive LN using CT and MRI, we think integration of information from all available imaging modalities is required to make more accurate clinical nodal staging so far. We have added this as study limitation.
Revision:
Line 300 on Page 10
“Fourth, we did not compare accuracy of clinical staging between specific imaging modality or criteria. So far, integrating information from all available imaging modalities is required to make more accurate clinical staging than relying on a single modality.”
References
- Loch, F.N.; Asbach, P.; Haas, M.; Seeliger, H.; Beyer, K.; Schineis, C.; Degro, C.E.; Margonis, G.A.; Kreis, M.E.; Kamphues, C. Accuracy of various criteria for lymph node staging in ductal adenocarcinoma of the pancreatic head by computed tomography and magnetic resonance imaging. World J. Surg. Oncol. 2020, 18, 213, doi:10.1186/s12957-020-01951-3.
- Nanashima, A.; Sakamoto, I.; Hayashi, T.; Tobinaga, S.; Araki, M.; Kunizaki, M.; Nonaka, T.; Takeshita, H.; Hidaka, S.; Sawai, T.; et al. Preoperative diagnosis of lymph node metastasis in biliary and pancreatic carcinomas: evaluation of the combination of multi-detector CT and serum CA19-9 level. Dig. Dis. Sci. 2010, 55, 3617-3626, doi:10.1007/s10620-010-1180-y.
- Soriano, A.; Castells, A.; Ayuso, C.; Ayuso, J.R.; de Caralt, M.T.; Gines, M.A.; Real, M.I.; Gilabert, R.; Quinto, L.; Trilla, A.; et al. Preoperative staging and tumor resectability assessment of pancreatic cancer: prospective study comparing endoscopic ultrasonography, helical computed tomography, magnetic resonance imaging, and angiography. Am. J. Gastroenterol 2004, 99, 492-501, doi:10.1111/j.1572-0241.2004.04087.x
- Unno, M.; Okumoto, T.; Katayose, Y.; Rikiyama, T.; Sato, A.; Motoi, F.; Oikawa, M.; Egawa, S.; Ishibashi, T. Preoperative assessment of hilar cholangiocarcinoma by multidetector row computed tomography. J. Hepatobiliary Pancreat. Surg. 2007, 14, 434-440, doi:10.1007/s00534-006-1191-4.
- Kim, Y.C.; Park, M.S.; Cha, S.W.; Chung, Y.E.; Lim, J.S.; Kim, K.S.; Kim, M.J.; Kim, K.W. Comparison of CT and MRI for presurgical characterization of paraaortic lymph nodes in patients with pancreatico-biliary carcinoma. World J. Gastroenterol. 2008, 14, 2208-2212.
- The authors comment that pathologic size was not an accurate predictor of OS. The suggest that adding information related to venous involvement may be more predictable. Is it possible to include this data from their database?
Answer:
Thank you for your valuable comment.
First, we did not comment pathologic size (by surgical specimen) was not accurate of survival prediction. We suggest that radiographic tumor size, particularly T3 criteria may not be adequate for clinical staging.
Our group published a paper regarding radiographic PV or SMV invasion that it could significantly affect the prognosis of patients with non-metastatic PC.1 Unfortunately, as we described in the study limitation, we are unable to suggest better T classification criteria when we added other factors such as venous invasion based on our database. The number of patients with resectable and borderline resectable PC might not be adequate for this and a larger multicenter study could suggest better clinical T staging system. We have added this in the discussion section.
Revision:
Line 305 on Page 10
“The number of patients with resectable and borderline resectable PC might not be adequate to discriminate minor prognostic differences between stages.”
References
- Kang, H.; Kim, S.S.; Sung, M.J.; Jo, J.H.; Lee, H.S.; Chung, M.J.; Park, J.Y.; Park, S.W.; Song, S.Y.; Park, M.S.; et al. Radiographic portal or superior mesenteric vein invasion is an independent prognostic factor in non-metastatic pancreatic ductal adenocarcinoma: A missing block of clinical T staging? Pancreatology 2020, 20, 952-959, doi:10.1016/j.pan.2020.05.017.
- The manuscript would be stronger if they excluded other types of cancer such as IPMN. Although they claim the prevalence is small, they should do a sensitivity analysis to confirm that it minimally influences their outcomes.
Answer:
Thank you for your valuable comment.
First, we correct “undifferentiated carcinoma” by “unspecified carcinoma”.
We performed same survival analysis in patients with confirmed adenocarcinoma after excluding patients with IPMN with invasive carcinoma and unspecified carcinoma. The results are consistent with those of all included patients, and we think this could strengthen our study.
However, as we described in study limitation, patients with IPMN with invasive carcinoma or unspecified carcinoma use same staging system and receive same treatment with adenocarcinoma. On the other hand, different staging system is used in patient with neuroendocrine tumor, and we excluded these patients from our study at patient selection. In this aspect, we suggest that the results from all included patients may be close enough to the real-world setting. The results from adenocarcinoma patients are presented in supplementary material 2 and 3, and we revised the manuscript as below.
Revision:
Line 80 on Page 2
“We excluded patients with confirmed extrapancreatic malignancy, and confirmed pancreatic neuroendocrine tumor or carcinoma, because different staging systems are used for these patients. We also excluded patients with or inadequate imaging data with which to conduct clinical staging.”
Table 2 on Page 4
Undifferentiated carcinoma -> Unspecified carcinoma
Line 183 on Page 6
“Same analysis performed in 942 patients with confirmed adenocarcinoma shows consistent results with those from all 1043 patients (Supplementary material 2).”
Line 196 on Page 7
“Same analysis performed in 456 patients with confirmed adenocarcinoma shows consistent results with those presented above (Supplementary material 3).”
Line 298 on Page 10
“In addition, we excluded patients with neuroendocrine tumors whose staging system and treatment differed from those of patients with adenocarcinoma.”
- You report that adjuvant therapy was given in 129 (12.4) patients. How is "adjuvant" therapy defined? Is it only after surgery or does it include those who received neoadjuvant therapy?
Answer:
Thank you for your valuable comment.
Adjuvant therapy in our study is defined as treatment aimed for prevention of recurrence and prolongation of survival after curative surgical resection. Therefore, all patient who had adjuvant therapy had undergone curative surgical resection before. Among them, 16 patients had been received neoadjuvant therapy before surgery.
If a patient experienced disease progression during neoadjuvant therapy and became having unresectable pancreatic cancer, the next treatment was recorded as palliative therapy.
- Can you include more information on what therapy patients received both in the neo-adjuvant and adjuvant settings?
Answer:
Thank you for your valuable comment.
We have analyzed detailed information of neoadjuvant and adjuvant therapy, and present as a table in Supplementary material 1. We also revised the manuscript as below.
Revision:
Line 144 on Page 4
“Detailed information of neoadjuvant and adjuvant therapy is presented in Supplementary material 1.”
Round 2
Reviewer 1 Report
I have no comments